# *Dicer1* Depletion Leads to DNA Damage Accumulation and Cell Death in a RET/PTC3 Papillary Thyroid Cancer Mouse Model, Thereby Inhibiting Tumor Progression

**DOI:** 10.3390/cells14181465

**Published:** 2025-09-19

**Authors:** Maria Rojo-Pardillo, Alice Augenlicht, Geneviève Dom, Jukka Kero, Bernard Robaye, Carine Maenhaut

**Affiliations:** 1IRIBHM J. E. Dumont, Université Libre de Bruxelles,1070 Brussels, Belgium; 2Department of Clinical Sciences, Faculty of Medicine, University of Turku, FI-20520 Turku, Finland; 3IRIBHM J. E. Dumont, Université Libre de Bruxelles, Campus Charleroi, 6041 Charleroi, Belgium; bernard.robaye@ulb.be

**Keywords:** thyroid, *Dicer1*, papillary thyroid cancer, RET/PTC3

## Abstract

Beyond well-known genetic drivers, microRNA dysregulation has emerged as a key contributor to thyroid tumorigenesis. Central to this process is *Dicer1*, a ribonuclease essential for microRNA maturation, whose expression is often reduced in papillary thyroid carcinoma (PTC). Evidence from previous studies suggest *Dicer1* functions as a context-dependent haplo-insufficient tumor suppressor gene: partial loss may promote tumor development, whereas complete loss may disrupt essential cellular functions, causing cell death and tumor suppression. However, the effects of partial or complete *Dicer1* loss in thyroid cancer remain unclear. To explore this, we genetically inactivated one (heterozygous) or both (homozygous) *Dicer1* alleles specifically in thyroid follicular cells of a RET/PTC3 transgenic mouse model using an inducible Cre-Lox system. Our findings deepen the current understanding of the RET/PTC3-driven PTC model by revealing an increased number of vimentin-positive cells and disruption in redox homeostasis. Additionally, whereas heterozygous *Dicer1* loss did not alter tumor progression in RET/PTC3 mice, total loss reduced tumor growth and led to accumulated DNA damage and cell death. These findings highlight the crucial role of *Dicer1* dosage in thyroid cancer progression and underscore its potential as a therapeutic target for aggressive PTC and other malignancies characterized by aberrant *Dicer1* expression.

## 1. Introduction

Thyroid follicular cell-derived tumors are the most prevalent endocrine malignancy and the sixth most common cancer among women worldwide, with an increasing incidence over recent decades [1,2,3,4]. This rise is attributed not only to improved detection and early diagnosis but also to environmental and lifestyle factors [1,3,4]. Thyroid neoplasms are clinically and biologically heterogeneous, ranging from slow-growing benign tumors to malignant tumors, further classified into three main groups: well-differentiated, poorly differentiated and undifferentiated carcinomas [3,5]. Papillary thyroid carcinoma (PTC), a well-differentiated carcinoma, is by far the most common subtype, accounting for up to 90% of thyroid cancer diagnoses [1,4]. Standard therapeutic approaches include surgery and radioactive iodine therapy, while advanced or refractory cases may require chemotherapy, targeted therapies, such as tyrosine kinase inhibitors, or immunotherapy [3,4,6]. Nevertheless, management differs significantly between pediatric and adult cases and should be adjusted accordingly [7]. Although PTC generally has a favorable prognosis, certain variants can be aggressive, with risks of persistence and recurrence [4], posing a significant clinical challenge and highlighting the importance of better understanding the molecular mechanisms influencing tumor aggressiveness.

The molecular landscape of PTC is defined by well-characterized driver mutations, predominantly involving alterations in the MAPK signaling pathway, specifically in the BRAF, RAS and RET/PTC genes, which are key contributors to tumor initiation and progression [8,9,10,11]. Nevertheless, in addition to these genetic drivers, other regulatory mechanisms are gaining recognition for their role in tumorigenesis such as microRNA dysregulation. MicroRNAs are small non-coding RNA molecules of approximately 20–25 nucleotides in length that regulate gene expression at the post-transcriptional level, by binding to target messenger RNAs, leading to either inhibition of translation or degradation of the mRNA [12]. Each microRNA can regulate multiple genes, and each gene can be targeted by distinct microRNAs [13,14], together constituting a highly intricate gene regulatory network that influences genes involved in a broad spectrum of biological processes, many of which are fundamental to cancer progression. Their balanced expression is essential for maintaining normal cellular function. Extensive evidence indicates that aberrant regulation of microRNAs is a characteristic feature of cancer [14,15,16], including thyroid cancer [17,18,19]. The mechanisms driving microRNA deregulation in cancer are not yet fully elucidated. However, defects in the biogenesis pathway, particularly involving key components like *Dicer1*, may contribute to these alterations.

*Dicer1* is an enzyme primarily known for its role in processing precursor microRNAs into their mature forms [20]. However, it also participates in the biogenesis and maturation of other non-coding RNAs, such as DNA damage response RNAs (ddRNAs) [21,22,23]. In several mouse *Dicer1* knockout studies, thyrocyte–specific deletion of *Dicer1* and impairment of miRNA synthesis have been shown to alter thyrocytes follicular organization, function and differentiation [24,25,26]. Mutations in *Dicer1*, although rare, have been shown to be implicated in cancer development [20,27]. Germline mutations in *Dicer1* cause *Dicer1* syndrome, a hereditary disorder that predisposes individuals to a variety of tumors, including thyroid neoplasms [28]. While germline mutations tend to be dispersed throughout the gene, somatic mutations are frequently concentrated in hotspots in exons 24 and 25, which encode the RNAseIIIb domain, part of the catalytic domain of *Dicer1* and essential for its proper function [20,29]. These mutations can disrupt normal microRNA processing and contribute to tumorigenesis. In addition to mutations, aberrant expression of *Dicer1* has also been observed in cancer [30,31,32,33,34]. Precisely, in PTC, *Dicer1* mRNA expression is often downregulated in both primary tumors and metastases compared to adjacent normal tissues and this downregulation has been associated with more aggressive tumor phenotypes, further suggesting that reduced *Dicer1* expression may contribute to tumor progression and malignancy [30].

Though *Dicer1* downregulation has been linked to tumor progression, complete loss of *Dicer1* function is extremely rare [20,24,35]. Previous studies suggest that *Dicer1* acts as a haplo-insufficient tumor suppressor gene. Partial loss of *Dicer1* contributes to genomic instability, by disrupting microRNA expression and impairing proper ddRNA production, thereby promoting tumorigenesis. On the other hand, complete loss of *Dicer1* leads to a collapse of gene expression regulation and DNA damage response, ultimately leading to cell death and inhibition of tumor development, as observed in models of lung cancer, sarcoma, retinoblastoma and prostate cancer [36,37,38,39]. However, this effect was not observed in lymphoma and angiosarcoma models [40,41], suggesting that the role of *Dicer1* as a haplo-insufficient tumor suppressor may be tissue-dependent, highlighting the complexity of its function in oncogenesis.

Previous studies have shown that thyrocyte-specific deletion of *Dicer1* alters follicular structure and function but is not sufficient to initiate tumor development [24,25,26]. Therefore, in this study, we evaluated the impact of partial or complete inactivation of *Dicer1* on thyroid cancer progression using a RET/PTC3 transgenic mouse model of PTC. To do so, we selectively inactivated one (*Dicer1*^+/−^) or both (*Dicer1*^−/−^) alleles specifically in thyroid follicular cells of two-month-old mice, a time point at which tumor initiation has already occurred and the model recapitulates key molecular and histopathological features of the human disease. We then assessed the consequences of *Dicer1* loss on tumor growth and differentiation.

## 2. Materials and Methods

### 2.1. Generation of the RET/PTC3 Dicer1^(−/−)^ or RET/PTC3 Dicer1^(+/−)^ Mice and Tissue and Blood Collection

The institutional ethics committee (CEBEA) approved the animal experiments performed in this study (N° 86, IRIBHM, LA 1500468, accepted on 5 May 2021, valid from 1 November 2015 to 1 November 2020, validity extended until 5 May 2026) and N° 30 Gos IBMM, LA 1500474, accepted on 5 June 2024, valid from 1 July 2024 to 30 June 2029). Animals were housed under standard laboratory conditions with free access to food and water, as well as appropriate care and monitoring to minimize pain and distress. Genotyping was performed from mouse digits (phalanges) using the following primers: RET/PTC3 Forward 5′-GGCCAGAGCCCTAAGGTGGGC-3′, Reverse 5′-AAGGGATTCAATTGCCATCCA-3′ (PCR product size 240 bp); Cre-ERT2 Forward 5′-ATGCCAACCTCACATTTCTTG-3′, Reverse 5′-AGTCCCTCACATCCTCAGGTT-3′ (PCR product size 480 bp); *Dicer1*: Forward 5′-CCTGACAGTGACGGTCCAAAG-3′, Reverse 5′-CATGACTCTTCAACTCAAACT-3′ (PCR product size 420 bp if floxed and 351 bp if not). 2-month-old *RET/PTC3* iTgCre-ERT2 *Dicer1*^(flox/flox)^ or ^(wt/flox)^ [42,43,44] mice were injected intraperitoneally for five consecutive days with vehicle (corn oil) with or without tamoxifen (75 mg/kg) and were sacrificed 1 day, 2 months or 6 months after tamoxifen treatment. Verification of *Dicer1* exon 24 recombination was performed using cDNA synthesized from RNA extracted from thyroid tissue, since genomic DNA could not be used because of material limitations, with primers Forward 5′-GTTTGACCATCCAGACGCAG-3′ and Reverse 5′-TTTCTCCTCATCCTCCTCG G-3′ (PCR product size 209 bp if recombined, 478 bp if not). Mice and thyroids were weighed and tissues were collected snap-frozen or preserved in paraffin or OCT after fixation in 4% paraformaldehyde, RNA extraction or histological analyses. Blood was collected and T4 serum concentrations were measured by ELISA (T4119AT, Calbiotech, El Cajon, CA, USA) following manufacturer’s instructions. Serum samples from hypothyroid mice lacking the DUOX maturation factors were used as controls [45].

### 2.2. RNA Extraction and Quantitative PCR Amplification

Total RNA was extracted from bulk mice thyroids using miRNeasy mini kit (2170004, Qiagen, Hilden, Germany) according to the manufacturer’s instructions. Total amount of RNA was quantified and treated with DNAse I (18068015, ThermoFisher Scientific, Waltham, MA, USA) and RNase Out (100000840, ThermoFisher Scientific, Waltham, MA, USA). mRNA reverse transcription was performed using Superscript II reverse transcription kit (18064022, ThermoFisher Scientific, Waltham, MA, USA) and quantitative PCR amplification was performed using KAPA SYBR FAST (KK4601, KapaBiosystems, Wilmington, MA, USA). qPCR primers are listed in Appendix A. *Tbp* and *Hprt* were used as internal normalizers for mRNA expression analysis [46]. The expression of each gene was normalized using the 2^−∆∆Ct^ method [47] and presented relative to controls (RET/PTC3 *Dicer1*^(+/+)^ thyroids).

### 2.3. Flow Cytometry Assays

Flow cytometry assays were performed on thyroid tissue that was enzymatically dissociated using collagenase (2 mg/mL, 30 min, 37 °C) and trypsin (2.5%, 5 min, 37 °C). APC/Cyanine7 anti-mouse CD326, EpCAM, (118218, BioLegend, San Diego, CA, USA) antibody was employed to identify and sort thyroid follicular cell population. Cell sorting was performed using Aria III from BD Biosciences, configured with 4 lasers (violet, blue, yellow-green, red). Data analysis was carried out using Diva software, version 9.

### 2.4. Histology and Immunofluorescence

Five micrometer-thick paraffin tissue sections were deparaffinized, rehydrated in an ethanol gradient and stained with hematoxylin-eosin. Paraffin or OCT slides were blocked in 1% BSA, 10% horse serum, 0.2% Triton PBS for 1 h at room temperature. Slides were incubated with the following primary antibodies diluted in blocking buffer in a humid chamber overnight at 4 °C: polyclonal rabbit anti-human thyroglobulin (Tg) 1/1000 (A0251, DAKO, Glostrup, Denmark), iodinated thyroglobulin (Tg-I) 1/500 (Provided by C. Ris-Stalpers, Laboratory of Pediatric Endocrinology, The Netherlands), thyroxine (T4) 1/500 (ORB11479, Biorbyt, Cambridge, UK), keratin type II/cytokeratin 8 (CK8) 1/500 (2EA, TROMA-I, Developmental Studies Hybridoma Bank, Iowa City, IA, USA), alpha-smooth muscle actin monoclonal antibody (1A4) eFluor™ 660 (α-SMA) 1/500 (50-9760-82, ThermoFisher Scientific, Waltham, MA, USA), anti-vimentin antibody [EPR3776] (VIM) 1/250 (AB92547, Abcam, Cambridge, UK), histone 2AX (p-Ser-139) 1/500 (NB100-384, Novus Biotechne, Centennial, CO, USA) or cleaved caspase-3 (Asp175, 5A1E) 1/500 (9664, Cell Signaling Technology, Danvers, MA, USA). Then, they were incubated for one hour with the following secondary antibodies diluted in blocking buffer and DAPI (4′,6-diamidino-2-phenylindole, dilactate, D3571, Invitrogen, Carlsbad, CA, USA) at room temperature: donkey anti-rabbit IgG (H+L) highly cross-adsorbed secondary antibody, Alexa Fluor™ 488, 1/250 (A-21206, ThermoFisher Scientific, Waltham, MA, USA), goat anti-rat IgG (H+L) cross-adsorbed secondary antibody, Alexa Fluor™ 546, 1/250 (A-11081, ThermoFisher Scientific, Waltham, MA, USA) or donkey anti-goat IgG (H+L) cross-adsorbed secondary antibody, Alexa Fluor™ 594, 1/250 (A-11058, ThermoFisher Scientific, Waltham, MA, USA). Slides were PBS-cleaned and mounted using mounting medium (C0563, DAKO, Glostrup, Denmark). Negative control samples (NC) were incubated only with the secondary antibody, omitting the primary antibody. Images were acquired at LiMiF http://limif.ulb.ac.be/ (accessed on 1 August 2025) on a Axio Observer Z1 inverted microscope. Images were quantified using Qupath 0.4.3.

### 2.5. TUNEL Staining for DNA Fragmentation Detection

DNA fragmentation detection was performed in OCT sections using the Click-iT™ Plus TUNEL Assay Kit (C10617, ThermoFisher Scientific, Waltham, MA, USA) according to the manufacturer’s protocol. To permeabilize the cells, sections were incubated with 0.1% Triton X-100 in sodium citrate 0.1 M (pH = 6) 0.1% for 2 min at room temperature. Positive controls were treated with DNase I in sodium citrate 0.1 M (pH = 6) for 10 min at room temperature to induce DNA breaks. Negative control samples were incubated only with the secondary antibody, omitting the TUNEL staining. After completing the TUNEL staining, the slides were stained as in Section 2.4.

### 2.6. Statistical Analyses

Statistical analyses were performed using Prism GraphPad 6.0. Data distribution was analyzed using the Shapiro–Wilk normality test. Statistically significant differences between two groups were determined using the Mann–Whitney *t*-test for non-parametric data or *t*-test for parametric data. Statistically significant differences for more than two groups were determined using Kruskal–Wallis test for non-parametric data or ANOVA for parametric data. The columns represent the mean values, and the error bars indicate the standard deviation (mean ± SD). * *p* < 0.05, ** *p* < 0.01 and *** *p* < 0.001.

## 3. Results

### 3.1. Cre-Mediated Recombination of Dicer1 in the Thyroid Is Confirmed

RET/PTC3 transgenic mice, a PTC model driven by the thyroid-specific expression of the RET/PTC3 oncogene under the thyroid-specific thyroglobulin promoter [43,48,49], were crossed with inducible Cre-ERT2 mice (iTgCre), where Cre-ERT2 expression was also controlled by the thyroglobulin promoter [40], carrying one or two alleles of *Dicer1* floxed at exon 24, which encodes part of the protein catalytic domain [41] (Figure 1A). Upon tamoxifen administration, Cre-ERT2 recognized and recombined the loxP sites, present in one or both alleles resulting in heterozygous or homozygous inactivation of *Dicer1* (RET/PTC3 *Dicer1*^(+/−)^ or RET/PTC3 *Dicer1*^(−/−)^ mice). The control group RET/PTC3 *Dicer1*^(+/+)^ was composed of RET/PTC3 Cre-ERT2 *Dicer1*^(flox/flox)^, ^(flox/wt)^ or ^(wt/wt)^ mice treated with the vehicle and RET/PTC3 *Dicer1*^(flox/flox)^, ^(flox/wt)^ or ^(wt/wt)^ treated with tamoxifen. This experimental design controls for any side effects from tamoxifen or the vehicle. Mice were injected with tamoxifen or vehicle at two months of age and sacrificed either 1 day, 2 months, or 6 months after treatment. These groups will hereafter be referred to as 2-, 4- and 8-month-old mice.

*Dicer1* recombination in the thyroid was confirmed by PCR on thyroid cDNA (Figure 1B,C). From 24 h after tamoxifen treatment, we found evidence of Cre-Lox recombination of *Dicer1*: the recombined allele (209 bp) was present in RET/PTC3 *Dicer1*^(−/−)^ and ^(+/−)^ mice thyroids. However, a small amount of recombined allele was also detected in the control thyroids, even in the absence of Cre, indicating some spontaneous recombination of LoxP sites (Figure 1C, RET/PTC3 *Dicer1*^(+/+)^, left panel). The recombined allele was never observed in WT mice or in mice lacking LoxP sites. On the other hand, the WT allele (478 bp) was present in the WT mice but also in RET/PTC3 *Dicer1*^(−/−)^, RET/PTC3 *Dicer1*^(+/−)^ and RET/PTC3 *Dicer1*^(+/+)^ mice. The presence of the WT allele can be attributed to the heterogeneity of the thyroid gland, where ∼30% of the cells are non-thyroidal [50] and not subjected to recombination, which, in this model, is strictly limited to thyroid follicular cells. An incomplete recombination of follicular cells could also be possible. To further analyze the origin of *Dicer1* remaining levels, we sorted the cells by flow cytometry and collected EpCAM+ cells, in order to identify epithelial cells, which, in the context of the thyroid, corresponds exclusively to follicular cells. We validated *Dicer1* recombination in follicular cells from RET/PTC3 *Dicer1*^(−/−)^ and RET/PTC3 *Dicer1*^(+/−)^ thyroids, where the recombined allele (209 bp) was detected (Figure 1C, right panel). On the other hand, the WT allele (478 bp) was present in the follicular cells from WT thyroids and RET/PTC3 *Dicer1*^(+/−)^ and RET/PTC3 *Dicer1*^(+/+)^ thyroid tumors, as expected, but also in RET/PTC3 *Dicer1*^(−/−)^ thyroid tumors. The detection of a WT band in the recombined samples signified the retention of exon 24 in our recombination model, suggesting the recombination was not complete.

Additionally, RET/PTC3 mRNA expression was assessed over time, as tumor dedifferentiation may lead to the loss or decreased expression of the oncogene. RET/PTC3 expression remained unchanged in 2-, 4- and 8-month-old mice (Figure 1D).

We further investigated *Dicer1* mRNA expression by RT-qPCR (Figure 2). First, we aimed to determine whether RET/PTC3 thyroid tumors exhibit downregulation of *Dicer1* mRNA expression compared to WT thyroids, as observed in human PTC [33,51]. Indeed, *Dicer1* mRNA levels were downregulated in thyroids from 4- and 8-month-old RET/PTC3 *Dicer1*^(+/+)^ thyroid tumors compared to WT thyroids (Figure 2A). The same tendency was observed in 2-month-old mice. This observation reinforces the suitability of our model for studying human PTC, as it closely mimics the molecular features of human PTC [48]. Then, we quantified the loss of exon 24 in *Dicer1* mRNA in RET/PTC3 *Dicer1*^(−/−)^ and RET/PTC3 *Dicer1*^(+/−)^ thyroid tumors compared to RET/PTC3 *Dicer1*^(+/+)^ mice tumors (Figure 2B). No differences in expression were observed between RET/PTC3 *Dicer1*^(+/−)^ and control RET/PTC3 *Dicer1*^(+/+)^ thyroid tumors, likely because both display reduced expression levels compared to normal thyroid. However, *Dicer1* exon 24 mRNA levels were markedly decreased in thyroid tumors from 2, 4- and 8-month-old RET/PTC3 *Dicer1*^(−/−)^ tumors.

### 3.2. Total Dicer1 Inactivation Leads to Reduced Thyroid Tumor Size

Concerning body weight, no differences were observed between groups: neither the presence of the tumor nor the inactivation of *Dicer1* impacted body weight (Figure 3A). In addition, thyroids were dissected from each animal and weighed. WT thyroids exhibited low tissue growth with no significant weight changes over time (Figure 3B). In contrast, thyroid tumors from RET/PTC3 *Dicer1*^(+/+)^ mice showed higher growth and continued to increase in size over time as expected [39,48].

Similar outcomes were observed in male and female groups. Although thyroid tumors weights from RET/PTC3 *Dicer1*^(+/+)^ and RET/PTC3 *Dicer1*^(+/−)^ mice were not significantly different, RET/PTC3 *Dicer1*^(−/−)^ tumors exhibited slightly slower growth curves and reduced tumor weight at 4 and 8 months of age (Figure 3B,C).

### 3.3. Although the Thyroid Follicular Structure Is Altered in Thyroid Tumors, Hormone Production Is Sustained

Histological modifications were observed in thyroid tumors compared to WT thyroids at all explored ages (Figure 4). However, no difference was observed when comparing the histology of RET/PTC3 *Dicer1*^(−/−)^, RET/PTC3 *Dicer1*^(+/−)^ and RET/PTC3 *Dicer1*^(+/+)^ thyroid tumors. RET/PTC3 tumors exhibited tissue dysplasia with histological changes, including large, hypertrophied follicles and increased number of non-follicular cells. Since hormone production depends on thyroid follicular structure, and since this structure is altered in thyroid tumors, we analyzed Tg, Tg-I and T4 presence by immunostaining as well as serum T4 levels by ELISA in 2-month-old mice. Immunostainings showed comparable Tg, Tg-I and T4 staining intensity and distribution in WT thyroids and RET/PTC3 thyroid tumors (Figure 5 and Figure 6). In the same way, the comparison of RET/PTC3 *Dicer1*^(−/−)^, RET/PTC3 *Dicer1*^(+/−)^ and RET/PTC3 *Dicer1*^(+/+)^ thyroid tumors revealed no differences in Tg, Tg-I or T4 staining. Consistently, T4 serum levels were comparable across RET/PTC3 *Dicer1*^(−/−)^, RET/PTC3 *Dicer1*^(+/−),^ RET/PTC3 *Dicer1*^(+/+)^ and WT mice (Appendix A). Similar findings were observed in 8-month-old mice.

### 3.4. RET/PTC3 Tumors Show Increased Proportion of Vimentin-Positive Cells, Regardless of Dicer1 Inactivation

Immunolabelings against cytokeratin-8 (CK8), vimentin and alpha-smooth muscle actin (α-SMA) were conducted to study the cellular composition of thyroid tumors (Figure 7). A significant increase in the number of vimentin-positive cells was observed in thyroid tumors from 2-, 4- and 8-month-old RET/PTC3 *Dicer1*^(+/+)^ mice compared to WT thyroids, which was expected given the tumoral context (Figure 7A,B and Appendix A). Vimentin-positive cells represented up to 40% of the thyroid tumor cell population. No significant difference was found between the thyroid tumors from RET/PTC3 *Dicer1*^(−/−)^, RET/PTC3 *Dicer1*^(+/−)^ and RET/PTC3 *Dicer1*^(+/+)^ mice.

Regarding α-SMA staining, no difference was observed in the percentages of α-SMA-positive cells between RET/PTC3 *Dicer1*^(+/+)^ tumors and WT thyroids in 2-, 4- and 8-month-old mice, regardless of *Dicer1* partial or total inactivation (Figure 7C and Appendix A). However, in 8-month-old mice thyroids, the number of alpha-SMA-positive cells was significantly greater in thyroids from RET/PTC3 *Dicer1*^(−/−)^ mice. This finding was not considered to have significant biological relevance, considering the small difference in the number of positive cells between groups and the dispersion of the samples (≈2–4%) (Appendix A).

### 3.5. Dicer1 Inactivation Does Not Affect the Altered Thyroid-Specific or Redox Homeostasis Gene Expression Patterns Observed in RET/PTC3 Thyroid Tumors

*Tg*, *Tpo*, *Nis*, *Tshr* and thyroid transcription factors *Pax8* and *Nkx2-1* are essential for thyroid function. We analyzed by RT-qPCR their mRNA expression in RET/PTC3 thyroid tumors and WT thyroids. Our findings indicated that *Tg*, *Nis* and *Tpo* mRNA expressions were downregulated in 2- (Figure 8) and 8-month-old (Appendix A) mice tumors compared to WT thyroids. This downregulation was maintained regardless of *Dicer1* inactivation. On the other hand, *Tshr*, *Pax8* and *Nkx2-1* mRNA expressions remained unaltered in thyroid tumors compared to WT thyroids in 2- and 8-month-old mice.

In parallel, we investigated the redox status of these tumors (Figure 8), given that oxidative stress is a hallmark of tumor metabolism [52]. *Ogg1* (8-Oxoguanine DNA Glycosylase), *Gpx2* (Glutathione Peroxidase 2) and *Nqo1* (NAD(P)H Quinone Dehydrogenase 1) are key enzymes involved in the cellular response to oxidative stress [53,54,55]. While *Ogg1* mRNA expression was not modified, we observed a downregulation of *Gpx2* and *Nqo1* mRNA expression in thyroid tumors from 2- and 8-month-old RET/PTC3 *Dicer1*^(+/+)^ mice compared to WT thyroids, suggesting that redox homeostasis is unbalanced in RET/PTC3 tumors. Partial or total loss of *Dicer1* did not impact this downregulation (Figure 8 and Appendix A).

### 3.6. Dicer1^(−/−)^ Tumor Cells Accumulate DNA Damage, Triggering Cell Death

The observed reduced weight (Figure 3B) of the RET/PTC3 *Dicer1*^(−/−)^ thyroid tumors could result from increased cell death, potentially through apoptosis. To investigate this, we analyzed the mRNA expression of the pro-apoptotic proteins Bim and Bax, and anti-apoptotic protein Bcl2, which are involved in early stages of apoptosis and performed TUNEL staining, which reflects DNA fragmentation, a hallmark of apoptosis. RT-qPCR analysis revealed an upregulation of *Bim* in RET/PTC3 *Dicer1*^(+/+)^ thyroid tumors compared to the WT thyroids from 2-month-old mice (Figure 9). Similarly, *Bim* mRNA expression levels were upregulated in RET/PTC3 *Dicer1*^(−/−)^ and RET/PTC3 *Dicer1*^(+/−)^ thyroid tumors, showing no impact of *Dicer1* inactivation. No difference was observed for *Bax* and *Bcl2* mRNA expression. The upregulation of *Bim* mRNA expression decreased over time, and by 8 months of age, *Bim* expression levels had returned to baseline in RET/PTC3 *Dicer1*^(+/+)^ and RET/PTC3 *Dicer1*^(+/−)^ thyroid tumors, reaching values comparable to those observed in the WT condition. However, in RET/PTC3 *Dicer1*^(−/−)^ thyroid tumors, *Bim* mRNA expression remained slighly upregulated.

Our data suggested increased apoptosis in tumors. Notably, in RET/PTC3 *Dicer1*^(−/−)^ mice, Bim upregulation was sustained over time and remained evident in 8-month-old. Hence, we further investigated DNA fragmentation in 2- and 4-month-old mice using the TUNEL (Terminal deoxynucleotidyl transferase dUTP nick end labeling) assay. TUNEL staining was first performed on thyroid sections from mice at 2 and 4 months of age. No differences were found between RET/PTC3 *Dicer1*^(+/+)^ tumors and WT mice, both showing very low percentages of TUNEL-positive cells that did not exceed 0.5% of the total cell population (Figure 10A). In the same way, no differences were found when comparing RET/PTC3 *Dicer1*^(−/−)^ or RET/PTC3 *Dicer1*^(+/−)^ thyroid tumors to RET/PTC3 *Dicer1*^(+/+)^ thyroid tumors in 2- and 4-month-old mice. Given that 2-month-old mice were sacrificed just one day after tamoxifen treatment, likely too early after *Dicer1* inactivation to detect DNA fragmentation, and that 4-month-old mice may represent a time point when dead cells have already been cleared, we chose to investigate an intermediate time point, i.e., 2.5-month-old mice. This intermediate stage was arbitrarily chosen and intended to balance the temporal gap between the early (2-month) and late (4-month) cohorts, thereby providing a more continuous view of the progression of molecular and cellular changes. At this time point, the amount of TUNEL-positive cells was significantly increased in RET/PTC3 *Dicer1*^(−/−)^ thyroid tumors (Figure 10A). Additionally, when analyzing the location of the TUNEL-positive cells through histological examination, we observed they were located within the follicles and were positive for CK8 staining, reaching up to 3.5% of the follicular cell population and supporting the idea that total inactivation of *Dicer1* in thyrocytes leads to cell death (Figure 10B,C).

We then performed cleaved caspase-3 staining, but we were unable to detect positive cells (Appendix A), suggesting that this cell death was caspase-3-independent and triggered by the activation of a non-apoptotic cell death pathway. To further investigate this process, we examined whether the TUNEL staining could result from DNA fragmentation associated with DNA damage rather than apoptosis. Specifically, we assessed γH2AX accumulation as a marker of DNA double-strand breaks and damage-induced signaling. Indeed, we observed increased γH2AX staining in RET/PTC3 *Dicer1*^(−/−)^ thyroid tumors, which corresponded with the elevated TUNEL staining, supporting the hypothesis that cell death is triggered by the accumulation of DNA damage (Figure 11).

## 4. Discussion

Aberrant expression of *Dicer1* contributes to a wide range of human pathologies, including cancer [30,31,32,33,34,56,57,58], making *Dicer1* an attractive target not only for cancer but also for other pathologies [56]. Reduced levels of *Dicer1* mRNA have been reported in tumors but total loss of *Dicer1* is very rare [20]. Previous studies have suggested that *Dicer1* functions as a haplo-insufficient tumor suppressor gene. However, emerging evidence indicates that this role may be context dependent. In our previous in vitro study, we investigated the impact of partial and complete *Dicer1* inactivation and found that total, but not partial, loss of *Dicer1* significantly impaired PTC derived cell lines behavior, leading to reduced proliferation, migration and invasion, along with increased cell death [51]. To explore this within an in vivo context, we induced Cre-mediated *Dicer1* inactivation in a murine model of PTC, RET/PTC3 transgenic mice, where *Dicer1* was partially ^(+/−)^ or totally ^(−/−)^ inactivated specifically in the thyroid through deletion of exon 24 [39,40,41]. Inactivation was induced by tamoxifen administration in two-month-old mice, as at this age, the model transiently mimics human PTC and shares several key features with the human disease [48,49].

Recombination of *Dicer1* exon 24 was confirmed in RET/PTC3 *Dicer1*^(−/−)^ and RET/PTC3 *Dicer1*^(+/−)^ mice thyroids. Surprisingly, traces of recombination were also observed in the control group RET/PTC3 *Dicer1*^(+/+)^. Previous analysis of the *Tg*Cre-ERT2 mice revealed that Cre-ERT2 mediated recombination occurred exclusively in thyrocytes, with no significant activity observed in controls, making Cre-ERT2 leakage unlikely in our model [40]. Unexpectedly, recombination of exon 24 was present in mice lacking Cre-ERT2 gene and Cre-ERT2 expression. This phenomenon could be explained by transient activation of Cre-ERT2 expression during early development as previously reported in other Cre-lox recombination models [59]. Although recombination was confirmed, partial inactivation of *Dicer1*^(+/−)^ did not reduce *Dicer1* expression beyond the levels observed in RET/PTC3 *Dicer1*^(+/+)^ controls, preventing us from drawing definitive conclusions about the effects of partial *Dicer1* loss. It remains unclear whether this lack of difference reflects a genuine absence of impact of partial *Dicer1* inactivation on tumor behavior or it reflects the fact that our heterozygous model and RET/PTC3 controls both display intermediate levels of *Dicer1* expression, thereby masking potential haplo-insufficiency effects. To overcome this limitation, it will be essential to identify alternative PTC models that do not show a reduction in *Dicer1* expression and subsequently investigate the effects of partial loss of *Dicer1* in these models. On the other hand, our RET/PTC3 *Dicer1*^(−/−)^ thyroid tumors showed markedly decreased expression of *Dicer1* exon 24 mRNA levels. However, residual expression persisted in EpCAM-positive-sorted thyroid follicular cells, indicating incomplete recombination. This incomplete recombination is a limitation of this study; it underscores the need to refine and optimize recombination techniques to achieve more efficient and complete *Dicer1* inactivation. Furthermore, it is essential to implement strategies that can differentiate cells with complete recombination from those that do not.

Powel et al. previously showed that body weight and thyroid hormone production do not differ between RET/PTC3-only and wild-type mice [39]. However, our model also includes the Cre-ERT2 gene and LoxP sites alongside *Dicer1* gene, in addition to the RET/PTC3 oncogene, which are designed to be neutral and do not affect the organism phenotype unless activated, making it probable that these mice behave like RET/PTC3-only mice. As expected, RET/PTC3 *Dicer1*^(+/+)^ mice behaved as RET-PTC3-only mice: we did not observe any differences in body weight or T4 production between RET/PTC3 *Dicer1*^(+/+)^ mice and WT mice and RET/PTC3 *Dicer1*^(+/+)^ tumors were significantly larger than WT thyroids [39,48,49]. Interestingly, when we compared tumor sizes between the different tumoral conditions, no significant differences were observed between thyroids from RET/PTC3 *Dicer1*^(+/−)^ and RET/PTC3 *Dicer1*^(+/+)^ mice. As mentioned before, these outcomes might be due to the constraints of this heterozygous model. However, we found that thyroid tumors from 4-month-old and 8-month-old RET/PTC3 *Dicer1*^(−/−)^ mice were significantly smaller than those from RET/PTC3 *Dicer1*^(+/−)^ and RET/PTC3 *Dicer1*^(+/+)^ mice, indicating that the complete loss of *Dicer1* leads to a reduction in tumor size. No differences in tumor size were observed in thyroid tumors from 2-month-old mice, likely due to the short period between tamoxifen administration and tumor measurement, as mice were sacrificed only one day after tamoxifen treatment.

According to previous data obtained by Powell et al., and Jin et al., on RET/PTC3-only mice, RET/PTC3 *Dicer1*^(+/+)^ mice thyroids exhibited tissue disorganization and dedifferentiation, characterized by aberrant follicular architecture, aberrant gene expression and an increased number of stromal cells [39,49]. Specifically, we observed a downregulation of thyroid specific genes such as *Tg*, *Nis* and *Tpo* mRNAs, reflecting the well-established effect of constitutive MAPK signaling activation, which is known to repress thyroid differentiation markers [60]. Additionally, our study provides additional insights, highlighting the dysregulation of redox balance in RET/PTC3 tumors. Precisely, *Gpx2* and *Nqo1* mRNAs, which code for proteins participating in redox homeostasis, were downregulated [52,53,54,55]. This downregulation likely contributes to the disruption of redox balance commonly observed in tumors, creating a pro-oxidant environment that promotes genomic instability and tumor progression [61]. Also, we observed an increase in the number of vimentin-positive cells, reaching up to 40% of the total population. To further assess the tumor microenvironment, we examined α-SMA expression, a marker of smooth muscle cells from blood vessel walls, myofibroblasts and cancer-associated fibroblasts (CAFs) [62] but found no differences across conditions in 2-month-old mice. Neither the histology, the thyroid-specific and oxidative gene expression profiles or the high percentages of vimentin-positive cells were impacted by *Dicer1* inactivation, demonstrating that these changes are attributable to the presence of the tumor itself and not influenced by *Dicer1* inactivation.

Since reduced tumor size in RET/PTC3 *Dicer1*^(−/−)^ thyroids may result from increased cell death, we first examined apoptosis, which converges on caspase 3/7 activation [63]. We analyzed the expression of key genes involved in apoptosis, including Bim, Bcl2 and Bax [64], whose balance determines cell fate. We found Bim mRNA upregulated in 2-month-old tumors regardless of *Dicer1* status, but only RET/PTC3 *Dicer1*^(−/−)^ thyroids maintained Bim upregulation at 8 months. Furthermore, TUNEL staining, which detect fragmented DNA [65], revealed a significant increase in TUNEL-positive cells in the thyroids of RET/PTC3 *Dicer1* ^(−/−)^ mice two weeks after tamoxifen treatment (i.e., at 2.5 months of age), reaching up to 3.5% of the follicular population, suggesting that complete *Dicer1* inactivation in thyroid cells may induce an irreversible cellular fate, culminating in cell death. Importantly, cleaved caspase-3 was not detected in any of the samples with TUNEL-positive cells, suggesting that cell death occurred through a caspase-3-independent mechanism. Notably, these TUNEL-positive cells consistently exhibited strong γH2AX staining, indicating the presence of unresolved DNA damage.

It is well-known that the thyroid relies on high levels of hydrogen peroxide (H_2_O_2_) for hormone synthesis, which can induce both single- and double-strand DNA breaks in thyrocytes, making it a potential mutagen for this organ [66,67]. Nevertheless, the thyroid is well equipped with antioxidant defenses to counteract potential damage, including enzymes such as *Gpx2*, which plays a critical role in neutralizing oxidative stress by reducing hydroperoxides, and *Nqo1*, which helps maintain redox balance by facilitating the detoxification of reactive quinones. Tumors are often characterized by elevated oxidative stress and increased ROS production, resulting from the deregulation of antioxidant enzymes. Consistently, the tumoral RET/PTC3 model exhibited downregulation of redox-related enzymes, including *Gpx2* and *Nqo1*, highlighting a compromised antioxidant defense and increasing susceptibility to oxidative stress and genomic instability. Interestingly, despite this redox imbalance, cells retaining complete or partial *Dicer1* expression were able to overcome this deficiency, resolve DNA damage, and maintain genomic integrity, at least to an extent sufficient to allow cell survival. In contrast, complete loss of *Dicer1* impaired the response to oxidative stress, leading to increased γH2AX phosphorylation, which precedes the DNA fragmentation detected by TUNEL staining, thereby suggesting a pathway from oxidative damage to cell death in *Dicer1*-deficient tumors (Figure 12). Although *Dicer1* is primarily known for its role in microRNA biogenesis, growing evidence shows that it is also involved in the production of other small non-coding RNAs, including DNA damage-induced ddRNAs, which play a major role in the DNA damage response (DDR), a process traditionally viewed as a signaling cascade mediated exclusively by proteins [21,22,23]. Literature suggests that without *Dicer1*, the production of ddRNAs is lost, compromising the DNA damage response and leading to the accumulation of DNA damage that ultimately drives cell death. Unfortunately, we were unable to provide clear evidence of ddRNA loss in *Dicer1*^(−/−)^ mice, as detecting these molecules remains technically challenging. Our results are in accordance with previous studies reporting the accumulation of DNA damage in mouse epidermis, cerebellum and medulloblastoma following *Dicer1* total deletion [68,69].

While our study could not definitively assess the potential haplo-insufficient tumor suppressor role of *Dicer1* due to limitations in evaluating partial loss, the effects observed upon complete *Dicer1* loss align well with Swahari hypothesis, which proposes that total loss of *Dicer1* results in impaired DNA damage response, accumulation of genomic instability and cell death, compromising tumor survival. They also align with earlier studies by Zhang et al., who reported that complete inactivation of *Dicer1* resulted in smaller tumors in a prostate cancer mouse model, with increased number of TUNEL-positive cells following *Dicer1* total loss [39] or Arrate et al., who found that *Dicer1* total loss was strongly disfavored during B cell lymphoma development [40]. These observations raise interest in considering full inactivation or loss of *Dicer1* as a potential therapeutic strategy for thyroid cancer and possibly other malignancies. While similar approaches have shown promising results in other pathological contexts [70], their success depends on the development of highly efficient, tissue-specific delivery systems to achieve complete *Dicer1* inactivation, which to date remain underdeveloped.

Future studies should investigate different timings of *Dicer1* inactivation as well as *Dicer1* inactivation in additional PTC mouse models and in other in vivo cancer models, particularly those where *Dicer1* expression is known to be dysregulated between tumors and normal tissues, to assess whether the effects observed in our study are consistent across different genetic backgrounds and to validate the relevance of our findings [71].

## 5. Conclusions

Our findings suggest that complete inactivation of *Dicer1* induces DNA damage accumulation and cell death in RET/PTC3-driven papillary thyroid carcinomas. In contrast, although recombination was verified, partial *Dicer1*^(+/−)^ inactivation did not further decrease expression relative to RET/PTC3 *Dicer1*^(+/+)^ controls, making it difficult to draw firm conclusions about its impact.

## Figures and Tables

**Figure 1 cells-14-01465-f001:**
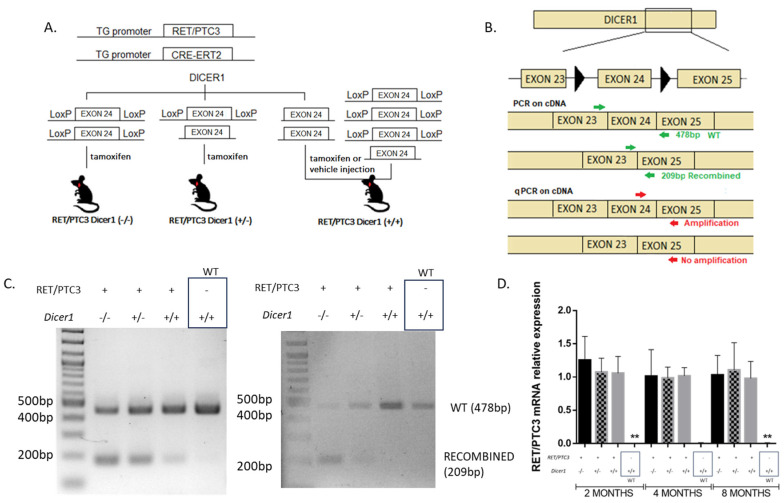
Generation and validation of a RET/PTC3 papillary thyroid carcinoma model with monoallelic or biallelic *Dicer1* deletion. (**A**) RET/PTC3 and *Cre-ERT2* expressions are thyroid-specific, regulated by the thyroglobulin promoter. *Dicer1* presents LoxP sites flanking exon 24 on one or both alleles. Following tamoxifen or vector injections, RET/PTC3 *Dicer1*^(−/−)^, RET/PTC3 *Dicer1*^(+/−)^ or RET/PTC3 *Dicer1*^(+/+)^ were generated. (**B**) *Dicer1* recombination was verified in the thyroid by PCR on cDNA using primers located in exons 23 and 25 (green arrows) and quantified by RT-qPCR using primers located in exons 24 and 25 (red arrows). (**C**) PCR products in 2-month-old mice (product size: WT 478 bp, recombined 209 bp) bulk thyroid tissue (**left**) or FACS-sorted epithelial population (**right**). (**D**) RT-qPCR analysis of RET/PTC3 mRNA expression in bulk thyroids of 2-month-old (RET/PTC3 *Dicer1*^(−/−)^ n = 7, RET/PTC3 *Dicer1*^(+/−)^ n = 10, RET/PTC3 *Dicer1*^(+/+)^ n = 10 and WT mice n = 8), 4-month-old (RET/PTC3 *Dicer1*^(−/−)^ n = 6, RET/PTC3 *Dicer1*^(+/−)^ n = 6, RET/PTC3 *Dicer1*^(+/+)^ n = 10 and WT mice n = 2) and 8-month-old ((RET/PTC3 *Dicer1*^(−/−)^ n = 7, RET/PTC3 *Dicer1*^(+/−)^ n = 9, RET/PTC3 *Dicer1*^(+/+)^ n = 8 and WT mice n = 6) mice. Statistically significant differences were determined using Kruskal–Wallis test. ** *p* < 0.01. The mean of each column was compared to the mean of the RET/PTC3 *Dicer1*^(*+/+*)^ column. The columns represent the mean values, and the error bars indicate the standard deviation (mean ± SD).

**Figure 2 cells-14-01465-f002:**
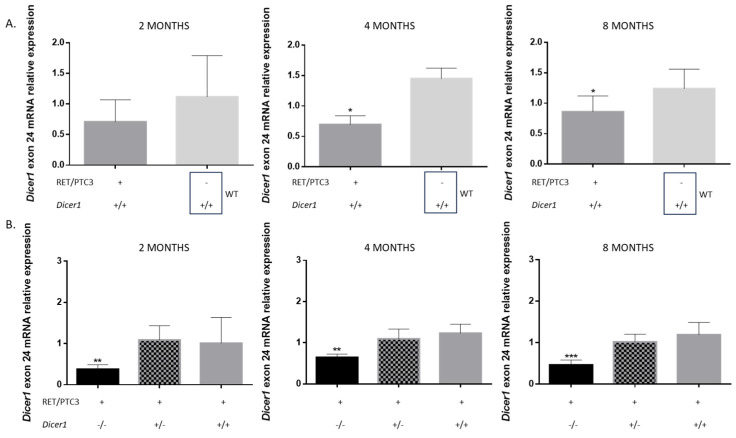
*Dicer1* exon 24 mRNA is downregulated in RET/PTC3 *Dicer1*^(+/+)^ and RET/PTC3 *Dicer1*^(+/−)^ PTC mouse models and further reduced in RET/PTC3 *Dicer1*^(−/−)^ mice. (**A**) RT-qPCR analysis of *Dicer1* exon 24 mRNA expression in bulk thyroids from 2-month-old (RET/PTC3 *Dicer1*^(+/+)^ n = 9, WT n = 5), 4-month-old (RET/PTC3 *Dicer1*^(+/+)^ n = 7, WT n = 3) and 8-month-old (RET/PTC3 *Dicer1*^(+/+)^ n = 8, WT n = 8) mice. Statistically significant differences were determined using Mann–Whitney *t*-test. * *p* < 0.05. (**B**) RT-qPCR analysis of *Dicer1* exon 24 mRNA expression in bulk thyroids of 2-month-old (RET/PTC3 *Dicer1*^(−/−)^ n = 7, RET/PTC3 *Dicer1*^(+/−)^ n = 9, RET/PTC3 *Dicer1*^(+/+)^ n = 10), 4-month-old (RET/PTC3 *Dicer1*^(−/−)^ n = 5, RET/PTC3 *Dicer1*^(+/−)^ n = 7, RET/PTC3 *Dicer1*^(+/+)^ n = 9) and 8-month-old (RET/PTC3 *Dicer1*^(−/−)^ n = 7, RET/PTC3 *Dicer1*^(+/−)^ n = 8, RET/PTC3 *Dicer1*^(+/+)^ n = 9) mice. Statistically significant differences were determined using Kruskal–Wallis test. ** *p* < 0.01 and *** *p* < 0.001. The mean of each column was compared to the mean of the RET/PTC3 column. The columns represent the mean values, and the error bars indicate the standard deviation (mean ± SD).

**Figure 3 cells-14-01465-f003:**
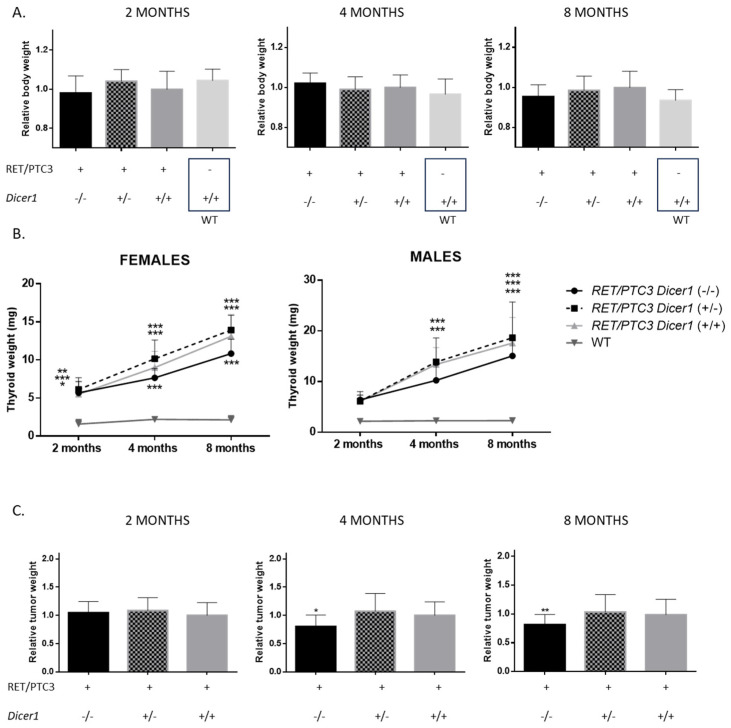
Total inactivation of *Dicer1* is associated with smaller thyroid tumors with stable body weight. (**A**) Body weight was normalized by the mean of the control group (females or males) (**B**) Thyroids were collected and weighed. Each point represents the mean of the group at that time point. Statistically significant differences were determined using two-way ANOVA test. * *p* < 0.05, ** *p* < 0.01 and *** *p* < 0.001. The mean of each column was compared to the mean of the WT column. (**C**) Thyroid weight was normalized by the mean of the thyroid weights of the control group (RET/PTC3 *Dicer1*^(+/+)^, females or males). Experiments were performed in 2-month-old (RET/PTC3 *Dicer1*^(−/−)^ n = 6, RET/PTC3 *Dicer1*^(+/−)^ n = 14, RET/PTC3 *Dicer1*^(+/+)^ n = 19, WT n = 12), 4-month-old (RET/PTC3 *Dicer1*^(−/−)^ n = 13, RET/PTC3 *Dicer1*^(+/−)^ n = 12, RET/PTC3 *Dicer1*^(+/+)^ n = 43, WT n = 15) and 8-month-old (RET/PTC3 *Dicer1*^(−/−)^ n = 25, RET/PTC3 *Dicer1*^(+/−)^ n = 32, RET/PTC3 *Dicer1*^(+/+)^ n = 45, WT n = 48) mice. Statistically significant differences were determined using Kruskal–Wallis test. * *p* < 0.05 and ** *p* < 0.01. The mean of each column was compared to the mean of the RET/PTC3 *Dicer1*^(+/+)^ column. The columns represent the mean values, and the error bars indicate the standard deviation (mean ± SD).

**Figure 4 cells-14-01465-f004:**
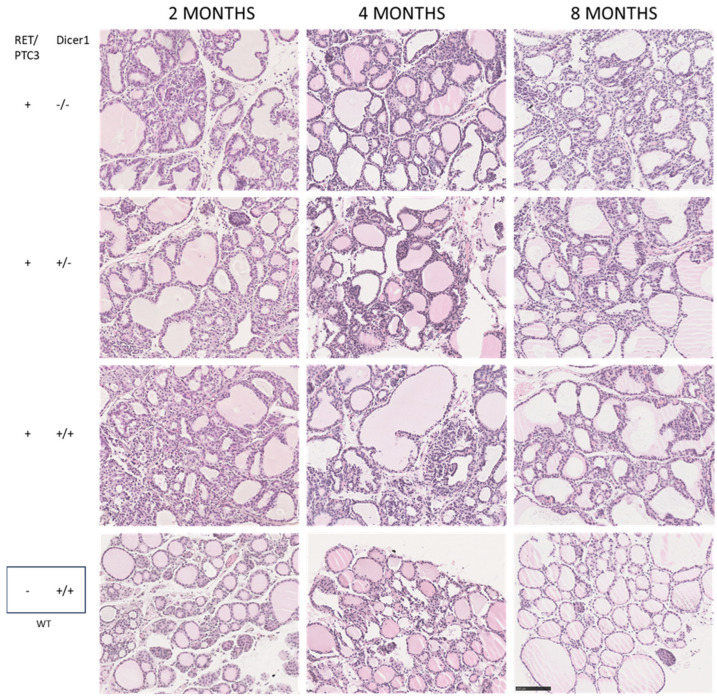
The thyroid follicular architecture is altered in RET/PTC3 *Dicer1*^(−/−)^, RET/PTC3 *Dicer1*^(+/−)^ and RET/PTC3 *Dicer1*^(+/+)^ thyroid tumors. Hematoxylin and eosin sections on paraffin slides from thyroids of 2-, 4- and 8-month-old RET/PTC3 *Dicer1*^(−/−)^, RET/PTC3 *Dicer1*^(+/−)^, RET/PTC3 *Dicer1*^(+/+)^ and WT mice. Images were captured at 20× magnification. Scale bar 100 µm.

**Figure 5 cells-14-01465-f005:**
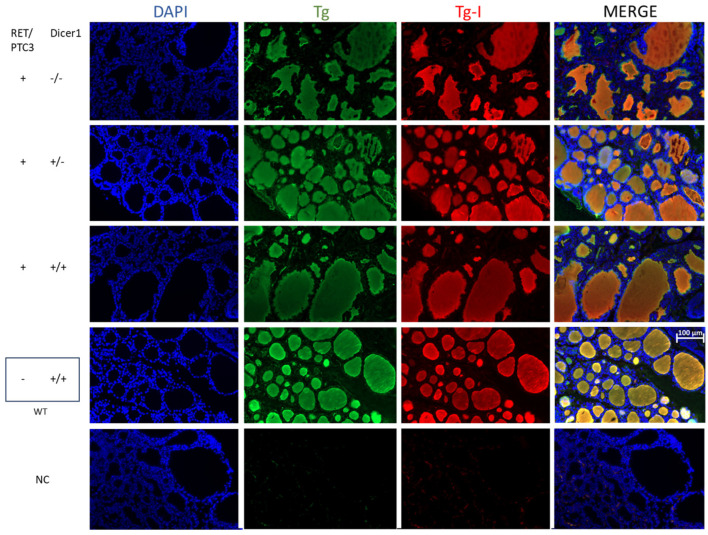
Tg and Tg-I stainings in thyroid tumors from RET/PTC3 *Dicer1*^(−/−)^, RET/PTC3 *Dicer1*^(+/−)^ and RET/PTC3 *Dicer1*^(+/+)^ mice are similar to WT thyroids despite the modified thyroid follicular architecture. DAPI staining (blue) and immunolabellings against thyroglobulin (green) and iodinated-thyroglobulin (red) in paraffin sections from representative thyroids from 2-month-old RET/PTC3 *Dicer1*^(−/−)^, RET/PTC3 *Dicer1*^(+/−)^, RET/PTC3 *Dicer1*^(+/+)^ and WT mice. Paraffin slides. NC: negative control (lack of primary antibody). Images were captured at 20× magnification. Scale bar 100 µm.

**Figure 6 cells-14-01465-f006:**
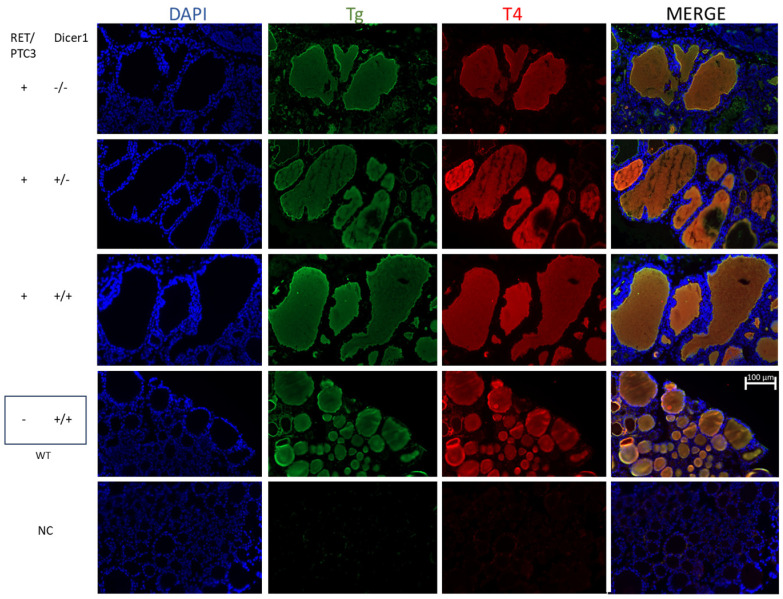
Tg and T4 staining patterns in thyroids tumors from RET/PTC3 *Dicer1*^(−/−)^, RET/PTC3 *Dicer1*^(+/−)^ and RET/PTC3 *Dicer1*^(+/+)^ mice remain comparable to those observed in WT thyroids. DAPI staining (blue) and immunolabellings against thyroglobulin (green) and T4 (red) in paraffin sections from representative thyroids from 2-month-old RET/PTC3 *Dicer1*^(−/−)^, RET/PTC3 *Dicer1*^(+/−)^, RET/PTC3 *Dicer1*^(+/+)^ and WT mice. Paraffin slides. NC: negative control (lack of primary antibody). Images were captured at 20× magnification. Scale bar 100 µm.

**Figure 7 cells-14-01465-f007:**
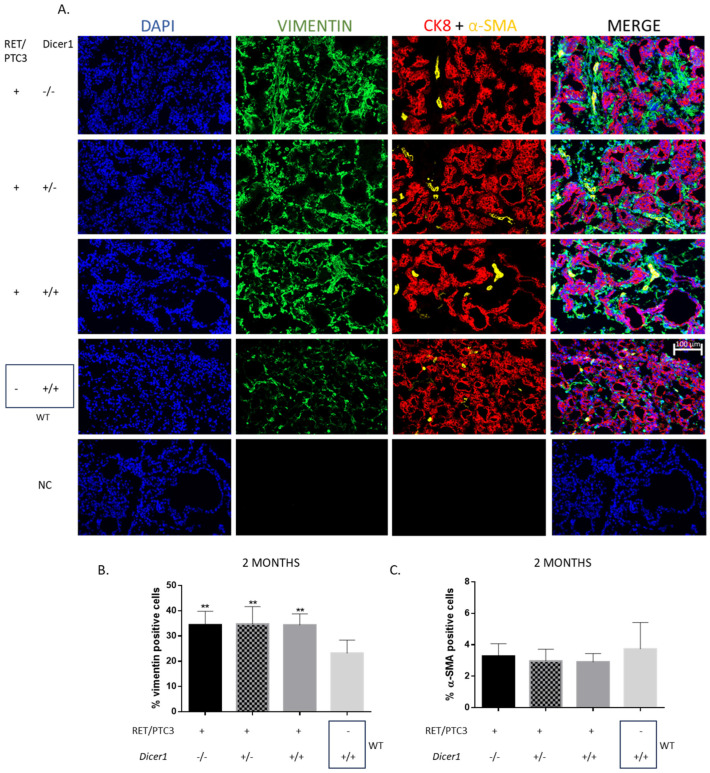
Vimentin expression is significantly increased in thyroid tumors, with vimentin-positive cells comprising as much as 40% of the total tumor cells, while α-SMA expression remains unaffected. (**A**) DAPI staining and immunolabellings against vimentin (green) cytokeratin-8 (CK8, red) and alpha-smooth muscle actin (α-SMA, yellow) in representative 2-month-old RET/PTC3 *Dicer1*^(−/−)^, RET/PTC3 *Dicer1*^(+/−)^, RET/PTC3 *Dicer1*^(+/+)^ and WT thyroids (OCT slides). NC: negative control (lack of primary antibody). Images were captured at 20× magnification. Scale bar 100 µm. (**B**) Quantification of the percentage of vimentin-positive cells (number of vimentin-positive cells/number of DAPI stained cells). (**C**) Quantification of the percentage of α-SMA-positive cells (number of α-SMA-positive cells/number of DAPI stained cells) in thyroids from 2-month-old (RET/PTC3 *Dicer1*^(−/−)^ n = 6, RET/PTC3 *Dicer1*^(+/−)^ n = 10, RET/PTC3 *Dicer1*^(+/+)^ n = 11, WT n = 5) mice. Statistically significant differences were determined using ANOVA test. ** *p* < 0.01. The mean of each column was compared to the mean of the WT column. The columns represent the mean values and the error bars indicate the standard deviation (mean ± SD).

**Figure 8 cells-14-01465-f008:**
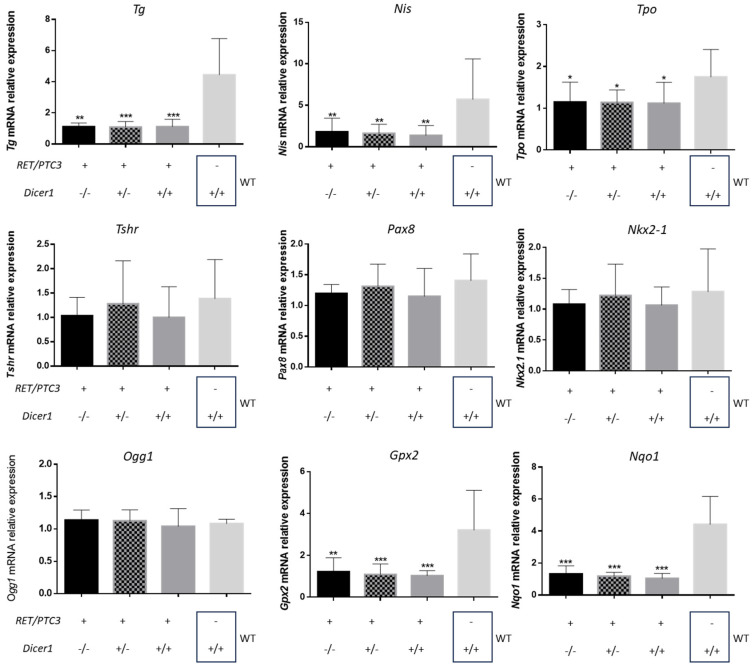
Altered expression of thyroid differentiation and redox genes in RET/PTC3-induced thyroid tumors from 2-month-old mice is not affected by partial or total loss of *Dicer1*. RT-qPCR analysis of mRNA expression of thyroglobulin (*Tg*), *Nis* (Na(+)/I(-) symporter), thyroid peroxidase (*Tpo*), thyroid stimulating hormone receptor (*Tshr*), Paired box 8 (*Pax8*), NK2 Homeobox 1 (*Nkx2-1*), 8-Oxoguanine DNA Glycosylase (*Ogg1*), Glutathione Peroxidase 2 (*Gpx2*) and NAD(P)H Quinone Dehydrogenase 1 (*Nqo1*) in thyroids of 2-month-old mice (RET/PTC3 *Dicer1*^(−/−)^ n = 7, RET/PTC3 *Dicer1*^(+/−)^ n = 9, RET/PTC3 *Dicer1*^(+/+)^ n = 10, WT n = 8). Statistically significant differences were determined using ANOVA test. * *p* < 0.05, ** *p* < 0.01 and *** *p* < 0.001. The mean of each column was compared to the mean of the WT column. The columns represent the mean values and the error bars indicate the standard deviation (mean ± SD).

**Figure 9 cells-14-01465-f009:**
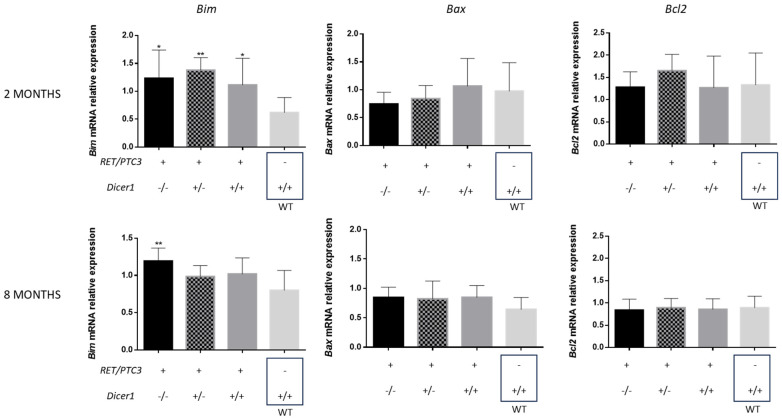
*Bim* but not *Bax* or *Bcl2* mRNA expression is upregulated in thyroid tumors. RT-qPCR analysis of mRNA expression of Bcl2 Like protein 11 (*Bim*), BCL2 Associated X (*Bax*) and Bcl2 Apoptosis Regulator (*Bcl2*) in thyroids from 2-month-old (RET/PTC3 *Dicer1*^(−/−)^ n = 7, RET/PTC3 *Dicer1*^(+/−)^ n = 9, RET/PTC3 *Dicer1*^(+/+)^ n = 10, WT n = 8) or 8 month-old (RET/PTC3 *Dicer1*^(−/−)^ n = 8, RET/PTC3 *Dicer1*^(+/−)^ n = 10, RET/PTC3 *Dicer1*^(+/+)^ n = 10, WT n = 8) mice. Statistically significant differences were determined using ANOVA test. * *p* < 0.05 and ** *p* < 0.01. The mean of each column was compared to the mean of the WT column. The columns represent the mean values and the error bars indicate the standard deviation (mean ± SD).

**Figure 10 cells-14-01465-f010:**
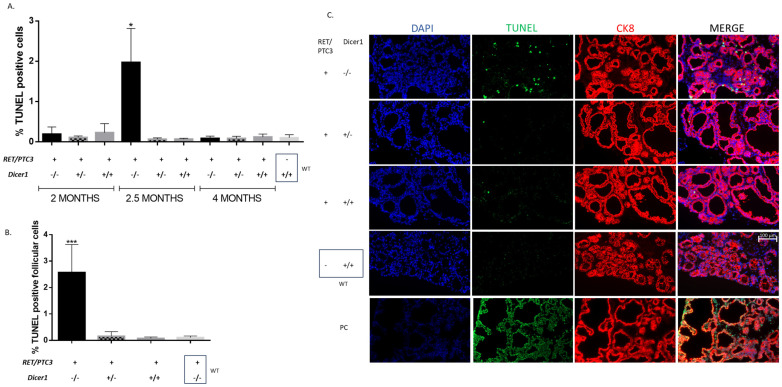
Total inactivation of *Dicer1* in thyroid tumors increases the number of TUNEL-positive cells. (**A**) Quantification of the percentage of TUNEL-positive cells (number of TUNEL-positive cells /number of DAPI stained cells) in thyroids from 2-month-old (RET/PTC3 *Dicer1*^(−/−)^ n = 5, RET/PTC3 *Dicer1*^(+/−)^ n = 5, RET/PTC3 *Dicer1*^(+/+)^ n = 5), 2.5-month-old (RET/PTC3 *Dicer1*^(−/−)^ n = 5, RET/PTC3 *Dicer1*^(+/−)^ n = 2, RET/PTC3 *Dicer1*^(+/+)^ n = 5) and 4-month-old (RET/PTC3 *Dicer1*^(−/−)^ n = 6, RET/PTC3 *Dicer1*^(+/−)^ n = 4, RET/PTC3 *Dicer1*^(+/+)^ n = 8) mice as well as in WT thyroids (n = 7). (**B**) Quantification of the percentage of TUNEL-positive follicular cells (number of TUNEL-positive cells /number of CK8-positive cells) in thyroids from 2.5-month-old (RET/PTC3 *Dicer1*^(−/−)^ n = 5, RET/PTC3 *Dicer1*^(+/−)^ n=2, RET/PTC3 *Dicer1*^(+/+)^ n = 5) and WT (n = 5) mice. (**C**) Example of DAPI staining and immunolabelling against cytokeratin-8 (CK8, red) and TUNEL (green) in thyroids from 2.5-month-old RET/PTC3 *Dicer1*^(−/−)^, RET/PTC3 *Dicer1*^(+/−)^, RET/PTC3 *Dicer1*^(+/+)^ and WT mice (OCT slides). PC: Positive control (DNAse treated cells). Images were captured at 20× magnification. Scale bar 100 µm. Statistically significant differences were determined using Kruskal–Wallis test. * *p* < 0.05 and *** *p* < 0.001 The mean of each column was compared to the mean of the WT column. The columns represent the mean values and the error bars indicate the standard deviation (mean ± SD).

**Figure 11 cells-14-01465-f011:**
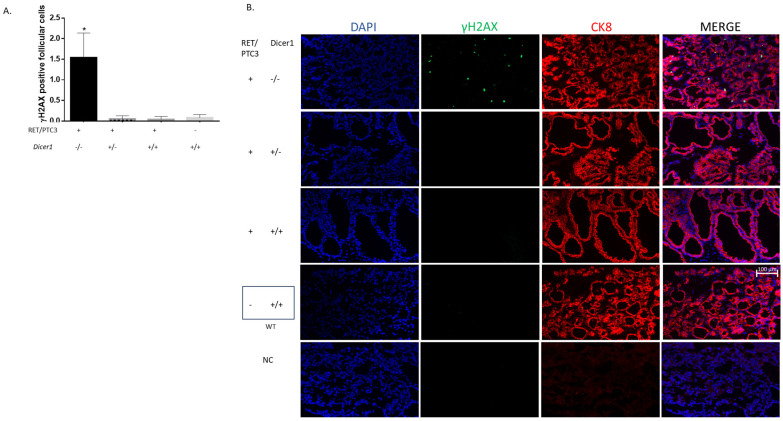
Total inactivation of *Dicer1* in thyroid tumors increases the number of γH2AX-positive cells. (**A**) Quantification of the percentage of γH2AX-positive cells (number of γH2AX-positive cells /number of CK8-positive cells) in thyroids from 2.5-month-old (RET/PTC3 *Dicer1*^(−/−)^ n = 5, RET/PTC3 *Dicer1*^(+/−)^ n = 2, RET/PTC3 *Dicer1*^(+/+)^ n = 5) and WT (n = 7) mice. Statistically significant differences were determined using Kruskal–Wallis test. * *p* < 0.05. The mean of each column was compared to the mean of the WT column. The columns represent the mean values and the error bars indicate the standard deviation (mean ± SD). (**B**) Example of DAPI staining and immunolabelling against γH2AX (green) and cytokeratin-8 (CK8, red) in thyroids from 2.5-month-old RET/PTC3 *Dicer1*^(−/−)^, RET/PTC3 *Dicer1*^(+/−)^, RET/PTC3 *Dicer1*^(+/+)^ and WT mice (OCT slides). NC: negative control (lack of primary antibody). Images were captured at 20× magnification. Scale bar 100 µm.

**Figure 12 cells-14-01465-f012:**
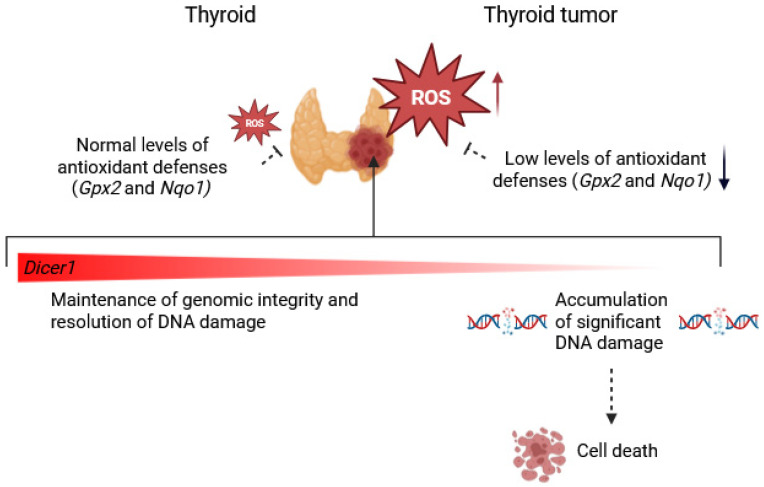
Schematic representation of oxidative stress control and cell fate in normal thyroids and RET/PTC3 thyroid tumors with or without *Dicer1*. While normal thyroids maintain genomic integrity through robust antioxidant defenses, including enzymes such as Gpx2 and Nqo1, RET/PTC3 tumors exhibit downregulation of these redox-related enzymes, compromising antioxidant defenses and increasing reactive oxygen species (ROS) levels, which can cause DNA damage. Cells retaining complete or partial *Dicer1* expression can resolve DNA damage and survive, whereas complete loss of *Dicer1* leads to cell death. Figure created with BioRender.com.

## Data Availability

The original contributions presented in this study are included in the article/Appendix A. Further inquiries can be directed to the corresponding authors.

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
