# Peer review of "Dicer1 Depletion Leads to DNA Damage Accumulation and Cell Death in a RET/PTC3 Papillary Thyroid Cancer Mouse Model, Thereby Inhibiting Tumor Progression"

_cells, 2025, doi:10.3390/cells14181465_

Round 1
Reviewer 1 Report
Comments and Suggestions for Authors
This manuscript presents a valuable in vivo investigation into the dosage-dependent role of Dicer1 in a RET/PTC3-driven model of papillary thyroid cancer. The study tackles a complex and relevant biological question, as Dicer1's function as both a haploinsufficient tumor suppressor and an essential gene for cell survival is highly context-dependent. The authors convincingly demonstrate that complete Dicer1 depletion, but not heterozygous loss, inhibits tumor growth. The strength of this work lies in the mechanistic data linking this tumor inhibition to an accumulation of DNA damage, evidenced by increased γH2AX and TUNEL staining, which ultimately leads to caspase-3-independent cell death. This provides a strong rationale for the observed phenotype and aligns with Dicer1's crucial role in the DNA damage response. However, the study's conclusions are constrained by a methodological limitation, which the authors acknowledge but perhaps understate in its impact. The Cre-Lox system employed resulted in incomplete recombination, with residual wild-type Dicer1 allele expression persisting even in the homozygous knockout group and evidence of spontaneous recombination in control animals. This technical issue critically impacts the interpretation of the data regarding partial Dicer1 loss. The finding that heterozygous Dicer1 depletion had no effect on tumor progression is difficult to interpret definitively, given that the control RET/PTC3 tumors already exhibit downregulated Dicer1 expression compared to wild-type thyroid tissue. This pre-existing downregulation may have masked any potential effects of haploinsufficiency, meaning the model, as executed, was not fully capable of testing this part of the central hypothesis. Furthermore, while the link between Dicer1 loss, DNA damage, and cell death is well-supported, the manuscript would benefit from a deeper exploration of the upstream mechanisms. The study reasonably posits that impaired redox homeostasis in the tumor contributes to the initial damage and that Dicer1's role in producing DNA damage response RNAs is compromised, leading to its accumulation. While demonstrating changes in ddRNA levels is technically challenging, the discussion could more thoroughly integrate the observed downregulation of redox enzymes like Gpx2 and Nqo1 into the proposed pathway from Dicer1 loss to cell death. The investigation of an intermediate 2.5-month time point was a crucial addition for capturing the peak of cell death, yet the rationale for this specific time point could be stated more explicitly in the results section. Additionally, the introduction section should be expanded with more recent references regarding the molecular biology, diagnosis, and histopathological features of thyroid carcinoma (10.1158/1055-9965.EPI-21-1440; doi: 10.1530/ETJ-22-0146; 10.3389/fendo.2023.1101410; 10.1002/cncy.22139; https://doi.org/10.1002/cncy.22224; 10.3390/diagnostics11061043).
Reviewer 2 Report
Comments and Suggestions for Authors
This is a very interesting study that described the effects of Dicer deletion in a model of thyroid cancer driven by RET/PTC3; It is a demanding study with animals that explored several aspects to understand the impact of Dicer deletion during thyroid oncogenesis. Please find below the comments. I think it can be amended, mostly rewritten and improved, for publication.
- Overall
- The title of the manuscript looks truncated, could it be rephrased?
- Methodologically I am not sure about the tamoxifen injection and the way the authors used to validate the deletion of exon 24. Why use cDNA instead of genomic DNA?
- The conclusion should be rewritten and be based on the results. It is too long and extrapolating to a potential application of Dicer inhibition.
- Abstract
- The abstract needs to be improved to mention after line 24 that this is a RET/PTC3 model
- Introduction
- Methods
- Line126: mice were injected with tamoxifen intraperitoneally or subcutaneously? Please specify
- Line 129: the genomic DNA used for validation of Dicer 1 knock-out was extracted from tail, ear or blood? Please specify
- Line 132: the fixation methods are missing for paraffin and OCT. Please, specify
- Line 144: please use the correct spelling for mouse genes in italic and the first letter in capital letter
- Results
- Line 224: why validate the Dicer deletion in cDNA and not in genomic DNA? Please explain and complement in the method section.
- Line 281: please remove this sentence that is methods “ Mice were sacrificed and their body weight was recorded. “
- Line 354: 3.5. Dicer1 inactivation does not impact the altered thyroid specific gene expression but disrupts redox homeostasis exhibited by the RET/PTC3 thyroid tumors. This title is wrong as the authors did not see any difference in dicer deletion in RET/PTC3 mice compared to RET/PTC3 Dicer wt. Please, rephrase this sentence
- Supplementary figure4: please correct the “exemple” in the figure for example
- Discussion
- The discussion is too long and should be refined as possible.
- How dicer deletion could impact in DNA damage and apoptosis adding to the reduction of redox status genes compared to other RET/PTC3 groups?
- Conclusion
- The conclusions should be based in the results obtained. The extrapolation about Dicer inhibition as therapy should be avoided, or should be attenuated. The conclusion is to long
Mainly in the discussion, it the text should be refined to avaoid at certain extent repetition of results
Round 2
Reviewer 1 Report
Comments and Suggestions for Authors
The authors have only partially addressed my comments. In fact, I would like them to consider all the suggested references because they provide broader background on the topic.
Author Response
We thank the reviewer for the suggestion. Four out of the five suggested references have been added to the Introduction section, as they indeed provide a broader background on the topic.
However, reference:
Dell'Aquila M, Gravina C, Cocomazzi A, Capodimonti S, Musarra T, Sfregola S, Fiorentino V, Revelli L, Martini M, Fadda G, Pantanowitz L, Larocca LM, Rossi ED. A large series of hyalinizing trabecular tumors: Cytomorphology and ancillary techniques on fine needle aspiration. Cancer Cytopathol. 2019 Jun;127(6):390-398. DOI:10.1002/cncy.22139. PMID: 31135104
focuses on hyalinizing trabecular tumors, which are not the topic of this paper. At present, we did not identify a clear connection of this reference to the background, and so it has not been incorporated. We would be grateful for any guidance on its intended placement.
Additionally, we have added another recent reference highlighting emerging technological advances in the field:
Tarabichi M, Demetter P, Craciun L, Maenhaut C, Detours V. Thyroid cancer under the scope of emerging technologies. Mol Cell Endocrinol. 2022 Feb 5;541:111491. doi:10.1016/j.mce.2021.111491. PMID: 34740746.
Furthermore, the entire bibliography has been carefully checked and corrected to ensure that all references appear correctly in the manuscript.
These updates address the comments from both the reviewers and the editors and further strengthen the focus and context of the Introduction.